# The prevalence of hypertension and hypertension control among married Namibian couples

**Alice Rose Weare**[1]*, **Zhixin Feng**[1,2], **Nuala McGrath**[1,3]

**1** CHERISH Programme, School of Primary Care, Population Sciences & Medical Education, Faculty of Medicine, University Hospital Southampton, Southampton, United Kingdom, **2** School of Geography and Planning, Sun Yat-sen University, Guangzhou, China, **3** Department of Social Statistics and Demography, Faculty of Social Sciences, University of Southampton, Southampton, United Kingdom

* alicerose.weare@gmail.com

**Data Availability Statement:** This was a retrospective study using third party data from the 2013 Namibian Demographic and Health Survey. DHS data are publicly available through the (https://

## Abstract

### Background

Previous studies suggest that having a marital partner with hypertension is associated with an individual's increased risk of hypertension, however this has not been investigated in sub-Saharan Africa despite hypertension being a common condition; the age-standardised prevalence of hypertension was 46.0% in 2013 in Namibia.

### Objective

To explore whether there is spousal concordance for hypertension and hypertension control in Namibia.

### Methods

Couples data from the 2013 Namibia Demographic and Health Survey were analysed. Bivariable and multivariable logistic regression models were used to explore the odds of individual's hypertension based on their partner's hypertension status, 492 couples. and the odds of hypertension control in individuals based on their partner's hypertension control (121 couples), where both members had hypertension. Separate models were built for female and male outcomes for both research questions to allow independent consideration of risk factors to be analysed for female and males.

### Results

The unadjusted odds ratio of 1.57 (CI 1.10–2.24) for hypertension among individuals (both sexes) whose partner had hypertension compared to those whose partner did not have hypertension, was attenuated to aOR 1.35 (CI 0.91–2.00) for females (after adjustment for age, BMI, diabetes, residence, individual and partner education) and aOR 1.42 (CI 0.98–2.07) for males (after adjustment for age and BMI). Females and males were significantly more likely to be in control of their hypertension if their partner also had controlled hypertension, aOR 3.69 (CI 1.23–11.12) and aOR 3.00 (CI 1.07–8.36) respectively.

dhsprogram.com/methodology/survey/survey-display-363.cfm).

**Funding:** This report is independent research supported by the National Institute for Health and Care Research using Official Development Assistance (ODA) funding (NIHR Global Health Research Professorship, Professor Nuala McGrath, RP-2017-08-ST2–008). ZF and NMcG were supported by this funding. The views expressed in this publication are those of the authors and not necessarily those of the NHS, the National Institute for Health and Care Research or the Department of Health and Social Care.

**Competing interests:** The authors have declared that no competing interests exist.

## Conclusions

Having a partner with hypertension was positively associated with having hypertension among married Namibian adults, although not statistically significant after adjustment. Partner's hypertension control was significantly associated with individual hypertension control. Couples—focused interventions, such as routine partner screening of hypertensive individuals, could be developed in Namibia.

## Introduction

Hypertension (high blood pressure) was the leading global risk factor for attributable deaths in a 2019 study of global burden of disease [1–3]. Hypertension is a major risk factor for cardiovascular and circulatory diseases, including stroke, myocardial infarction and renal failure [4]. In 2019, 9.3% (95% CI 8.2–10.5) of disease adjusted life years (DALYs) worldwide were attributable to hypertension [3].

There is evidence to suggest that having a marital partner with hypertension increases one's risk of the condition (spousal concordance), a meta-analysis of eight studies (from the UK, USA, Brazil and Russia) found a positive association of hypertension status between spouses in every study [5–8]. However, there is a gap in this research for Namibia and the rest of sub-Saharan Africa (SSA). There is also little research into spousal concordance for hypertension control among couples where both partners have hypertension. High spousal concordance for health risk behaviours, such as physical exercise and diet, suggests that there could be benefits of couples-focused interventions for a condition with significant modifiable risk factors like hypertension [9, 10]. Interdependence theory suggests that individuals who undergo a transition in motivation for health behaviour change from an individual-focus towards relationship-focus are more likely to support their partner's health [11, 12]. This promotes communal coping, a theory based on the idea that a partner's health risk is viewed by the couple as 'our problem' not 'your problem' [12]. The benefits of communal coping could apply to couple's health behaviours as part of the management of the risk factors for hypertension, as well as a shared approach towards controlling hypertension [11].

Namibia is one of the largest and most sparsely populated countries in SSA, with an estimated population of 2.45 million, with a life expectancy at birth of 64 years for males and females combined in 2019 [13, 14]. The 2013 Namibia Demographic and Health Survey (DHS) found that the age-standardised prevalence of hypertension was 46.0% in adults aged 35–64 [15]. The country has no national health insurance scheme and over 80% of adults aged 15–49 rely on the government funded health system [14]. SSA, including Namibia, faces the increasing burden of hypertension, with poor rates of awareness, treatment and control of hypertension in a generally uninsured population [14, 16]. We aimed to explore spousal concordance for hypertension and hypertension control in Namibia, using the 2013 Namibia DHS.

### Research questions

1. Is there spousal concordance in hypertension status?

2. In couples where both partners have hypertension, is there spousal concordance in hypertension control?

## Methods

### Study setting and data collection

This study was a secondary analysis of the latest Namibian DHS, which took place in 2013. The DHS programme aims to provide demographic and health data for policymaking and national health programmes [14]. This was the first national survey in Namibia to collect biomarker data, including blood pressure (BP) readings. The DHS final report provides details of data collection, the training of data collectors, the real time quality assurance of data from supervisors and the post survey quality assurance [14].

### Sample design and weight

The DHS used a two-stage stratified cluster design in order to conduct nationally representative household surveys [14]. In brief, the first stage involved selecting 554 enumeration areas (or clusters) (269 in urban clusters and 285 in rural clusters) with a stratified probability proportional to size selection using the sampling frame of the 2011 Namibia Population and Housing Census [14]. In the second stage, 20 households were selected in every urban and rural cluster according to equal probability systematic sampling [14].

Sampling weights were required for analysis of the DHS data to ensure the representativeness at a national level, given the study design, and variations in response rates to different components of the survey [14]. The DHS individual men's weight was used for the weighted analyses, following standard DHS advice for couple's analyses [17].

### Questionnaires

There were three DHS questionnaires administered: household, men's and women's [14]. The household questionnaire was administered in all selected households, and the individual women's questionnaire was administered to all females aged 15–49 years in selected households [14]. In half of the selected households all males, aged 15–64 years, were invited to complete an individual men's questionnaire [14]. Among the same half of selected households, the household questionnaire also included biomarker questions for all eligible males and females aged between 35–64 years. Alongside the biomarker questions eligible individuals were asked for consent to measure their BP. For this paper, the male and female data sets were merged to identify couples in which both partners had completed the survey questionnaire. Analyses were limited to couples in which both partners were aged between 35–64 years and had their BP recorded and a CONSORT diagram of sample selection for analyses was created (Fig 1).

### Ethics

The 2013 Namibian DHS questionnaires and procedures were reviewed and approved by the ICF Institutional Review Board and the Ministry of Health and Social Services Biomedical Research Committee. All participants gave written consent prior to taking part in the DHS questionnaires and having their blood pressure measured [14], All data were fully anonymized by the DHS program before we accessed them for our study [14]. Ethics approval for our study was granted by the University of Southampton Ethics and Research Governance Online (ERGO) committee.

### Outcome variables

We created a binary variable for hypertensive status (Y/N) using the World Health Organisation (WHO) hypertension classification of a systolic reading (SBP) $\geq140$ mmHg and/or a diastolic BP reading (DBP) $\geq90$ mmHg or individuals on antihypertensive medication with a 'normal' reading (SBP <120–139 mmHg and a DBP 80–89 mmHg), this was consistent with

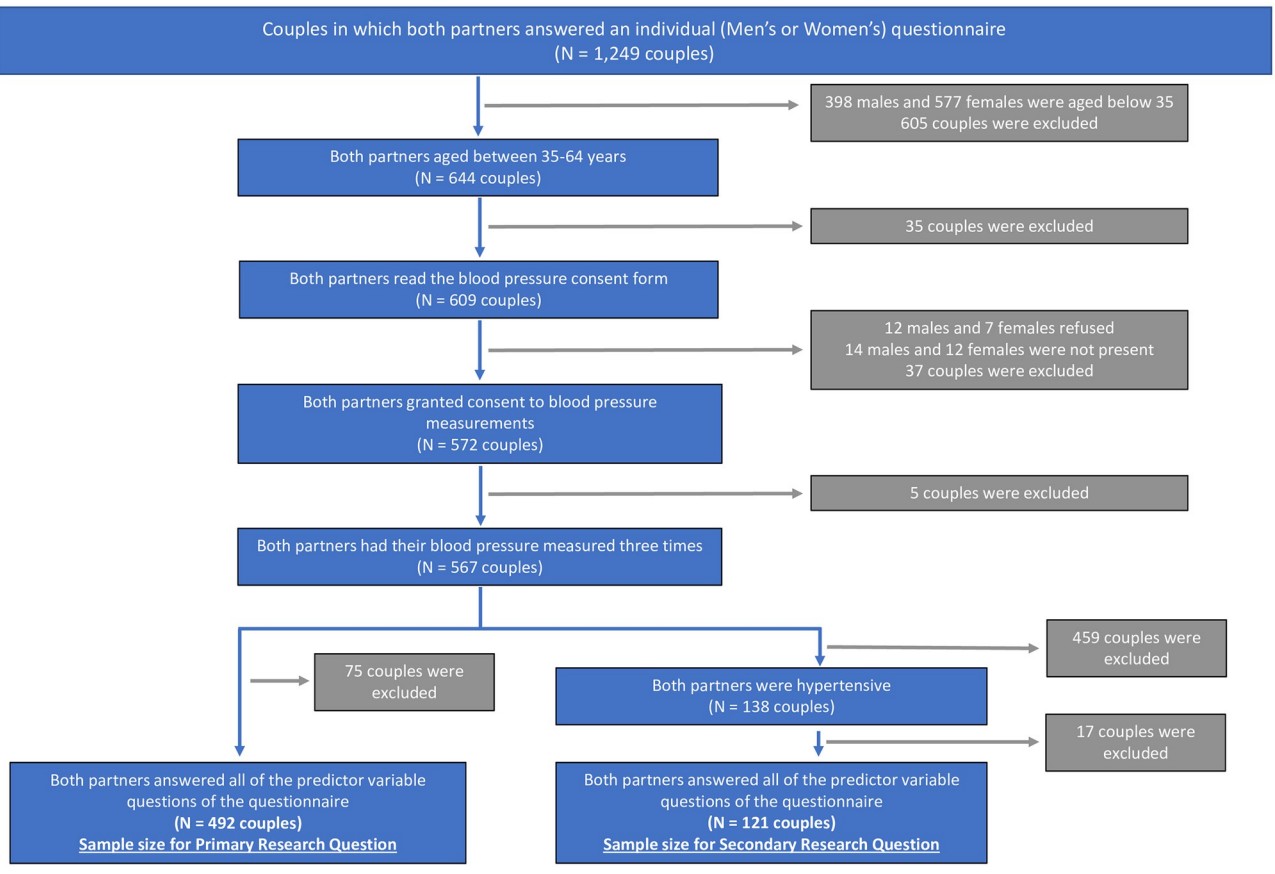

**Fig 1. A CONSORT diagram of sample selection from the 2013 Namibian DHS participants.**

the 2013 DHS report [14]. According to the WHO, hypertension diagnosis requires an individual's BP to be elevated on two different days, however the Namibian DHS recorded three BP measurements on one day and used an average of the second and third measurements. Therefore, within this study, hypertension does not necessarily mean a clinical diagnosis; instead, it is used as an indication of prevalence in the population at the time of the survey [14].

Similarly, following DHS operationalisation of hypertension control using average BP measurements and antihypertensive medication self-report [14], a binary variable was created to categorise each hypertensive individual as having their hypertension 'Controlled' or 'Uncontrolled'. Individuals were asked 'Have you ever been told by a doctor or other health worker that you have high blood pressure or hypertension?' [14], those that responded 'Yes' were defined as 'Aware' of their hypertension and the 'No' group were defined as 'Unaware' if they had elevated blood pressure. The Uncontrolled category included hypertensive individuals who were either 'Unaware' or those who were 'Aware' but not controlled (i.e., had elevated blood pressure at the time of survey). The 'Controlled' category was defined as individuals who were 'Aware' but did not have elevated blood pressure at the time of survey.

## Hypertension risk factor variables

Key hypertension risk factors identified from the literature [15] and available in the dataset were considered in the model for each research question: (age [16, 18], obesity [19],

education [20], diabetes status [21, 22], current smoking status [23–25]) for each individual partner and at the couples level: household wealth [6, 20, 26] and urban vs rural residence [27]. Age was considered in the model as a binary indicator representing 35–49 years vs 50–64 years, as was residence (urban vs rural) [14]. Height (m) and weight (kg) of participants were used to calculate their body mass index (BMI) (kg/m2) and then grouped into the WHO categories of Underweight (BMI<18.5), Normal (18.5–24.9), Overweight (25–29.9) and Obese (≥30) [14]. For smoking status, current smoking status was considered in models as a binary indicator (Yes vs No). Using the DHS definition, an individual was classified as having diabetes if he/she had a fasting plasma glucose of >7 mmol/L or was currently taking diabetes medication, diabetes status was grouped into 'No Diabetes' and 'Have Diabetes' [14].

Education was defined by the individual's highest level of education attainment at the time of survey and considered in the models as a categorical variable using dummy indicators for 'No education', 'Primary', 'Secondary" and 'More than secondary' [14]. Wealth was categorised by the DHS wealth quintile calculations of wealth factors including household assets, into 'Poorest', 'Poorer', 'Middle', 'Richer' and 'Richest' [14]. Adjustment for further known hypertension risk factors such as physical inactivity (PA) [15], alcohol consumption [15] and salt intake [15, 28] were beyond the scope of the study as these data were not collected in this survey (PA and salt intake) or only collected for a subset of the analysis sample (alcohol consumption) [14]

## Data analysis

Descriptive weighted analysis of the individual characteristics of male and female partners among the couples included in the analysis for each research question was conducted (Tables 1 and 4), and the Pearson chi-squared test was then used to test the associations between the outcome variable for each research question and each characteristic.

We built separate logistic regression models for female and male outcomes for both research questions, to allow consideration of risk factors to be analysed independently for female and male outcomes, with the primary exposure of interest in each model being the partner's hypertension status / control status, as in previous studies of spousal concordance [6–8, 26].

Potential additions were generally considered in the model as binary variables due to the small sample size leading to a limited power to explore associations using categorical variables. The reduction of categorical variables into binary indicators was driven by the distributions of those variables in Tables 1 and 4, these variables included index education, index BMI, partner's education and partner's BMI, with the binary indicator definition not necessarily kept the same for the index vs partner variables (Tables 2–6). For example, in Table 2, the female education binary indicator was defined as 'at least primary education' vs 'no education' whereas their partner's education binary indicator was 'higher education' vs 'secondary education or less'.

Data analysis was guided by previous published hypertension analyses in which separate models were built for female and male outcomes [6–8, 26]. We built bivariable models to examine the association of each potential factor with each outcome, always including the partner hypertension status/control variable as our primary exposure of interest (referred to in this paper as bivariable Models). Age-adjusted bivariable models are reported separately in the tables, to isolate the impact of adjusting for age on our association of interest from adjustment for all individual level factors (Age-adjusted Bivariable Model). To build multivariable models for each outcome, we first considered individual level factors (Model A). We then considered the addition of shared couples' factors (Model B), and the contribution of individual partner factors (Model C). A final parsimonious multivariable model was also run; including only

**Table 1. Proportion of females and males with a hypertensive status (based on individual and partner's characteristics).**

| | Females | | | Males | | |
|---|---|---|---|---|---|---|
| | Unweighted N (492) | Weighted N (416.5) | Weighted Percentage of Females with Hypertensive Status | Unweighted N (492) | Weighted N (416.5) | Weighted Percentage of Males with Hypertensive Status |
| **Individual Characteristics** | | | | | | |
| **Age** | | | *** | | | *** |
| 35–49 | 341 | 285.1 | 38.0% | 263 | 217.2 | 38.4% |
| 50–64 | 151 | 131.3 | 57.7% | 229 | 199.2 | 64.8% |
| **Education** | | | | | | |
| No education | 66 | 50.8 | 52.0% | 93 | 73.2 | 45.9% |
| Primary | 156 | 133.2 | 44.9% | 145 | 118.9 | 51.7% |
| Secondary | 220 | 181.7 | 43.4% | 206 | 180.1 | 50.8% |
| Higher | 50 | 50.7 | 37.6% | 48 | 44.2 | 58.4% |
| **BMI status** | | | *** | | | *** |
| Underweight | 31 | 28.3 | 23.9% | 62 | 54.6 | 25.6% |
| Normal | 166 | 161.1 | 36.5% | 235 | 202.6 | 43.5% |
| Overweight | 124 | 92.5 | 43% | 117 | 101.6 | 67.9% |
| Obese | 171 | 134.6 | 58.6% | 78 | 57.6 | 71.7% |
| **Current smoking status** | | | ** | | | |
| Don't smoke | 418 | 367.7 | 43.1% | 350 | 305.7 | 49.5% |
| Do smoke | 74 | 48.7 | 56.8% | 142 | 110.7 | 55.2% |
| **Diabetes status[1]** | | | *** | | | ** |
| No Diabetes | 458 | 391.5 | 41.8% | 449 | 381.9 | 50% |
| Have Diabetes | 34 | 25.0 | 82.3% | 43 | 34.6 | 67.0% |
| **Couples Level Characteristics** | | | | | | |
| **Residence** | | | *** | | | ** |
| Urban | 257 | 225.0 | 53.1% | 257 | 225.0 | 57.0% |
| Rural | 235 | 191.4 | 33.7% | 235 | 191.4 | 44.1% |
| **Household wealth quintiles** | | | ** | | | ** |
| Poorest | 77 | 72.3 | 24.8% | 77 | 72.3 | 32.2% |
| Poorer | 68 | 58.9 | 56.0% | 68 | 58.9 | 55.2% |
| Middle | 86 | 38.5 | 70.8% | 86 | 38.5 | 48.1% |
| Richer | 113 | 92.0 | 50.3% | 113 | 92.0 | 48.8% |
| Richest | 148 | 122.6 | 48.7% | 148 | 122.6 | 63.5% |
| **Partner's Characteristics** | | | | | | |
| **Partner's Age** | | | ** | | | *** |
| 35–49 | 263 | 217.2 | 39.1% | 341 | 285.1 | 45.1% |
| 50–64 | 229 | 199.2 | 49.8% | 151 | 131.3 | 63.9% |
| **Partner's Education** | | | | | | ** |
| No education | 99 | 73.2 | 45.9% | 66 | 50.8 | 44.5% |
| Primary | 145 | 118.9 | 38.5% | 156 | 133.2 | 47.5% |
| Secondary | 206 | 180.1 | 50.6% | 220 | 181.7 | 49.7% |
| Higher | 48 | 44.2 | 29.5% | 50 | 50.7 | 71.5% |
| **Partner's BMI status** | | | | | | ** |
| Underweight | 62 | 54.6 | 41.8% | 31 | 28.3 | 31.0% |
| Normal | 235 | 202.6 | 40.2% | 166 | 161.1 | 45.7% |
| Overweight | 117 | 101.6 | 56.5% | 124 | 92.5 | 53.0% |
| Obese | 78 | 57.6 | 39.5% | 171 | 134.6 | 60.2% |

*(Continued)*

**Table 1.** (Continued)

| | Females | | | Males | | |
|---|---|---|---|---|---|---|
| | Unweighted N (492) | Weighted N (416.5) | Weighted Percentage of Females with Hypertensive Status | Unweighted N (492) | Weighted N (416.5) | Weighted Percentage of Males with Hypertensive Status |
| Partner's Current Smoking status | | | ** | | | * |
| Don't smoke | 350 | 305.7 | 43.1% | 418 | 367.7 | 52.5% |
| Do smoke | 142 | 110.7 | 47.4% | 74 | 48.7 | 40.0% |
| Partner's Diabetes status[1] | | | * | | | * |
| No Diabetes | 449 | 381.9 | 41.8% | 458 | 391.5 | 50% |
| Have Diabetes | 43 | 34.6 | 47.0% | 34 | 25.0 | 67.0% |
| | | | | | | Total |
| | 492 | 416.5 | 45.1% | 492 | 416.5 | 51.0 |

Pearson Chi—Square—

***P < 0.01,

**P < 0.05,

*P < 0.1

[1]—An individual was classified as living with diabetes (Y/N) if they had a fasting plasma glucose of 7 mmol/L or above at the time of the survey. Or the individual had a 'normal' fasting plasma glucose, below 7 mmol/L, at the time of the survey and was on medication to control their diabetes.

variables that contributed significantly at the 5% level using a likelihood ratio test, and our primary exposure variable (Final Multivariable Model).

# Results

## Hypertension prevalence based on individual and partner's characteristics

The weighted results in Table 1 estimate that 51.0% of males and 45.1% of females are living with hypertension. We observe similar patterns in the individual characteristics for males and females, for example, with older adults (50–64 years) having higher prevalence of hypertension than younger age adults (35–49 years), $p<0.01$ for both sexes. Hypertension prevalence was higher in individuals living with diabetes than those not living with diabetes ($p<0.01$ for females and $p = 0.04$ for males). Both sexes had an increasing hypertension prevalence with increasing BMI, $p<0.01$. Those living in urban residence had higher prevalence of hypertension than those living in rural areas, $p<0.01$ for both sexes.

Patterns were less similar for males and female hypertension prevalence across partner characteristics. In general, male hypertension prevalence differed significantly across levels of all the female partner characteristics considered, while female hypertension prevalence remained similar across levels of the male partner characteristics except for partner age ($p = 0.04$) and partner BMI ($p = 0.05$).

## Spousal concordance in hypertension status

Tables 2 and 3 present the unadjusted and adjusted results of logistic regression models for the odds of hypertension for females and males respectively. Both males and females were significantly more likely to have hypertension if their partner was also hypertensive, OR 1.57 (CI 1.10–2.24), $p = 0.01$ (bivariable models in Tables 2 and 3). Female hypertension status was no longer statistically significantly associated with their partner's hypertension status in the final parsimonious multivariable model aOR 1.35 (CI 0.91–2.00), $p = 0.14$, after

**Table 2. Odds of female hypertension (bivariable and multivariable logistic regression models) (N = 492).**

| Factors considered | Bivariable Models adjusted for male hypertension status[1] | Bivariable Model (adjusting for Female Age) | Multivariable Model A (adjusting for female factors) | Multivariable Model B (Model A adjusting for couple factors) | Multivariable Model C (Model B adjusting for male factors) | Final Multivariable Model [3] |
|---|---|---|---|---|---|---|
| **Male partner's Hypertensive Status (ref: no-Hypertension)** | **1.57 (1.10–2.24)\*\*** | **1.44 (1.00–2.07)\*** | **1.42 (0.97–2.09)\*** | **1.36 (0.92–2.01)** | **1.37 (0.90–2.06)** | **1.35 (0.91–2.00)** |
| **Female's Characteristics** | | | | | | |
| 50–64 years (ref: 35–49) | 2.14 (1.44–3.17)\*\*\* | 2.14 (1.4–3.17)\*\*\* | 1.79 (1.18–2.73)\*\*\* | 1.96 (1.27–3.03)\*\*\* | 1.89 (1.12–3.17)\*\* | 2.18 (1.41–3.38)\*\*\* |
| At least primary education[4] (ref: No education) | 0.58 (0.34–0.99)\* | | 0.59 (0.36–1.05) | 0.49 (0.27–0.89)\*\* | 0.53 (0.30–0.97)\*\* | 0.51 (0.28–0.91)\*\* |
| BMI[5] (ref: Underweight / Normal) | | | | | | |
| Overweight | 1.75 (1.10–2.79)\*\* | | 1.59 (0.98–2.60)\* | 1.40 (0.84–2.32) | 1.41 (0.84–2.36) | 1.44 (0.87–2.39) |
| Obese | 2.95 (1.92–4.52)\*\*\* | | 2.88 (1.83–4.51)\*\*\* | 2.54 (1.58–4.09)\*\*\* | 2.87 (1.74–4.72)\*\*\* | 2.76 (1.74–4.37)\*\*\* |
| Current smoker (last 24hrs) (ref: Doesn't smoke) | 1.80 (1.09–2.98)\*\* | | 1.67 (0.97–2.85)\* | 1.56 (0.91–2.68) | 1.35 (0.76–2.41) | |
| Living with Diabetes[2] (ref: Not living with Diabetes) | 7.74 (2.93–20.41)\*\*\* | | 6.68 (2.47–18.05)\*\*\* | 6.77 (2.47–18.55)\*\*\* | 7.41 (2.64–20.8)\*\*\* | 7.41 (2.66–20.6)\*\*\* |
| **Couples Level Characteristics** | | | | | | |
| Rural Residence (ref: Urban) | 0.54 (0.38–0.78)\*\*\* | | | 0.53 (0.34–0.80)\*\*\* | 0.48 (0.31–0.76)\*\*\* | 0.46 (0.31–0.69)\*\*\* |
| Greater household wealth[6] (ref: poorest) | 2.37 (1.38–4.07)\*\*\* | | | 1.24 (0.65–2.37) | 1.34 (0.69–2.59) | |
| **Male Partner's Characteristics** | | | | | | |
| 50–64 (ref: 35–49) | 1.56 (1.08–2.25)\*\* | | | | 1.15 (0.71–1.87) | |
| Higher Education[7] (ref: No education / Primary/ Secondary) | 0.51 (0.27–0.97)\*\* | | | | 0.37 (0.17–0.77)\*\*\* | 0.35 (0.17–0.72)\*\*\* |
| BMI[8] Overweight / Obese (ref: Underweight /Normal) | 1.01 (0.07–1.47) | | | | 0.74 (0.47–1.16) | |
| Current smokes (last 24hrs) (ref: Not a current smoker) | 1.48 (1.00–2.20)\*\* | | | | 1.08 (0.68–1.73) | |
| Living with Diabetes[2] (ref: Not living with Diabetes) | 1.65 (0.87–3.13) | | | | 1.55 (0.75–3.18) | |

\*\*\*P < 0.01,

\*\*P < 0.05,

\*P < 0.1

[1]—Bivariable models—adjusting for partner hypertension status (key association of interest) + relevant factor for that row

[2]—An individual was classified as living with diabetes (Y/N) if they had a fasting plasma glucose of 7 mmol/L or above at the time of the survey. Or the individual had a 'normal' fasting plasma glucose, below 7 mmol/L, at the time of the survey and was on medication to control their diabetes.

[3]—Final multivariable model—adjusting for partner hypertension status (key association of interest) + variables that contributed significantly at the 5% level to the models, using a likelihood ratio test

[4]—'At least primary' created by combining primary, secondary and higher education categories

[5]—'BMI'—reference created by combining underweight and normal BMI categories

[6]—'Greater household wealth'—created by combining the poorer, middle, richer and richest DHS wealth quintiles

[7]—Male partner 'Higher Education'—reference created by combining no education, primary and secondary categories

[8]—Male partner 'Overweight / Obese'—created by combining overweight and obese BMI categories (for the reference underweight and normal weight BMI categories were combined)

adjustment for female age, education, BMI, diabetes, residence and partner's education (Table 2, final column). For the final male multivariable model (Table 3, final column), there was borderline significance for the association between male hypertension status and their partner's hypertension status, aOR 1.42 (CI 0.98–2.07), p = 0.07, after adjustment for male age and BMI.

### Spousal concordance in hypertension control

Table 4 presents the weighted percentage of hypertensive individuals with controlled hypertension based on individual and partner characteristics, for females and males respectively. Among the 121 hypertensive couples in the sample, 21 females and 25 males had controlled hypertension. The weighted percentage of hypertensive individuals with controlled hypertension was 18.6% and 18.1%, for females and males respectively. In both genders, the groups of hypertensive individuals with the greatest proportion with controlled hypertension are those with a higher level of education, those living with diabetes and those with a partner with controlled hypertension. Forty percent of females with a partner whose hypertension was controlled also had controlled hypertension, this was 38.8% for males, Table 4.

Tables 5 and 6 present the unadjusted and adjusted results of logistic regression models for the odds of controlled hypertension for females and males respectively. Individuals were significantly more likely to be in control of their hypertension if their partner was also in control of their hypertension, OR 3.00 (CI 1.08–8.36) p = 0.04, in unadjusted models (Tables 5 and 6). Female hypertension control remained statistically significantly associated with their partner's hypertension control in the final parsimonious multivariable model, aOR 3.69 (CI 1.23–11.12), p = 0.02, after adjustment for both female BMI and male partner BMI, (Table 5, final column). There were no variables that added significantly (p<0.1) to a model with female partner hypertension control status (key association of interest) for male hypertension control (Table 6, final column).

## Discussion

### Spousal concordance in hypertension status

This was the first study to explore spousal concordance in hypertension status and hypertension control among Namibian couples, aged 35–64 years. In our analyses, partner hypertension was significantly associated with individual hypertension in unadjusted models (OR 1.57 (CI 1.10–2.24), Tables 2 and 3) and the estimate of this association was only slightly attenuated in adjusted models, however it was no longer statistically significant (female aOR 1.35 (CI 0.91–2.00), Table 2 and male aOR 1.42 (0.98–2.07), Table 3). These results are consistent with and of a similar effect size to results from a meta-analysis of spousal concordance for hypertension in other regions of the world by Wang et al., which found that having a spouse with hypertension significantly increased an individual's risk of hypertension (male and female combined) by 41% (aOR 1.41 CI 1.21–1.64) [5].

Age, BMI and smoking status were reported to be important risk factors for hypertension in past studies of spousal concordance for hypertension [7, 8]. Our final model for female hypertension found individual and partner education level, individual diabetes status and urban residence to be significantly associated with increased odds of hypertension, as well as age and BMI, while smoking status did not remain in the final model. Our final model for male hypertension found individual age and individual BMI were significantly associated with increased odds of hypertension but smoking status was not.

Our findings for significant hypertension risk factors among these Namibian couples are generally consistent with previous literature from other parts of the world [15, 16, 18, 19].

**Table 3. Odds of male hypertension (bivariable and multivariable logistic regression models) (N = 492).**

| Factors considered | Bivariable Models adjusted for female hypertension status[1] | Bivariable Model (adjusting for male age) | Multivariable Model A (adjusting for male factors) | Multivariable Model B (Model A adjusting for couple factors) | Multivariable Model C (Model B adjusting for female factors) | Final Multivariable Model [3] |
|---|---|---|---|---|---|---|
| **Female partner's Hypertensive Status (ref: no Hypertension)** | **1.57 (1.10–2.24)**\*\* | **1.43 (0.99–2.06)** \* | **1.40 (0.96–2.04)**\* | **1.34 (0.91–1.97)** | **1.35 (0.90–2.01)** | **1.42 (0.98–2.07)** \* |
| **Male's Characteristics** | | | | | | |
| 50–64 years (ref: 35–49) | 2.30 (1.59–3.31)\*\*\* | 2.30 (1.59–3.31) \*\*\* | 2.26 (1.54–3.32)\*\*\* | 2.26 (1.53–3.34)\*\*\* | 2.18 (1.38–3.44)\*\*\* | 2.41 (1.66–3.51) \*\*\* |
| Secondary / Higher Education[4] (ref: No education / Primary) | 0.97 (0.68–1.39) | | 0.78 (0.51–1.17) | 0.73 (0.48–1.12) | 0.68 (0.44–1.07) | |
| Overweight / Obese[5] (ref: Underweight / Normal) | 2.34 (1.61–3.39)\*\*\* | | 2.62 (1.73–3.96)\*\*\* | 2.49 (1.64–3.80)\*\*\* | 2.42 (1.58–3.70)\*\*\* | 2.46 (1.68–3.60) \*\*\* |
| Current smoker (last 24hrs) (ref: Doesn't smoke) | 1.01 (0.68–1.50) | | 1.09 (0.72–1.66) | 1.09 (0.72–1.65) | 1.21 (0.77–1.90) | |
| Living with Diabetes[2] (ref: Not living with Diabetes) | 2.02 (1.04–3.89)\*\* | | 1.66 (0.83–3.36) | 1.62 (0.80–3.28) | 1.59 (0.78–3.24) | |
| **Couples Level Characteristics** | | | | | | |
| Rural Residence (ref: Urban) | 0.83 (0.58–1.19)\*\* | | | 0.90 (0.58–1.38) | 0.90 (0.58–1.39) | |
| Greater household wealth[6] (ref: poorest) | 1.77 (1.06–2.96)\*\* | | | 1.29 (0.72–2.31) | 1.26 (0.70–2.28) | |
| **Female Partner's Characteristics** | | | | | | |
| 50–64 (ref: 35–49) | 1.73 (1.16–2.57)\*\*\* | | | | 1.12 (0.68–1.85) | |
| At least primary education[7] (ref: No education) | 1.31 (0.77–2.22) | | | | 1.25 (0.69–2.28) | |
| Normal / Overweight / Obese[8] (ref: Underweight) | 0.59 (0.28–1.26) | | | | 0.80 (0.35–1.79) | |
| Current smokes (last 24hrs) (ref: Not a current smoker) | 0.73 (0.44–1.21) | | | | 0.71 (0.40–1.26) | |
| Living with Diabetes[2] (ref: Not living with Diabetes) | 1.32 (0.64–2.72) | | | | 1.11 (0.51–2.42) | |

\*\*\*P < 0.01,

\*\*P < 0.05,

\*P < 0.1

[1]—Bivariable models—adjusting for partner hypertension status (key association of interest) + relevant factor for that row

[2]—An individual was classified as living with diabetes (Y/N) if they had a fasting plasma glucose of 7 mmol/L or above at the time of the survey. Or the individual had a 'normal' fasting plasma glucose, below 7 mmol/L, at the time of the survey and was on medication to control their diabetes.

[3]—Final multivariable model—adjusting for partner hypertension status (key association of interest) + variables that contributed significantly at the 5% level to the models, using a likelihood ratio test

[4]—'Secondary / Higher Education' created by combining secondary and higher education categories (for the reference no education and primary were combined)

[5]—'Overweight / Obese' created by combining overweight and obese BMI categories (for the reference underweight and normal weight BMI categories were combined)

[6]—'Greater household wealth' created by combining the poorer, middle, richer and richest DHS wealth quintiles

[7]—Female partner 'At least primary' created by combining primary, secondary and higher education categories

[8]—Female partner 'Normal / Overweight / Obese' created by combining normal weight, overweight and obese BMI categories

**Table 4. Proportion of females and males with controlled hypertension (based on individual and partner's characteristics among couples where both partners were living with hypertension).**

| | Females | | | Male | | |
|---|---|---|---|---|---|---|
| | Unweighted N (121) | Weighted N (106.7) | Weighted Percentage of N with Controlled Hypertension | Unweighted N (121) | Weighted N (106.7) | Weighted Percentage of N with Controlled Hypertension |
| **Individual Control** | | | | | | |
| Not Controlled | 96 | 86.8 | 0.0% | 100 | 87.4 | 0.0% |
| Controlled | 25 | 19.9 | 100% | 21 | 19.3 | 100% |
| **Partner's Control** | | | | | | |
| Not Controlled | 100 | 87.4 | 13.9% | 96 | 86.8 | 13.3% |
| Controlled | 21 | 19.3 | 40.0% | 25 | 19.9 | 38.8% |
| **Individual Characteristics** | | | | | | |
| **Age** | | | | | | |
| 35–49 | 67 | 56.4 | 17.8% | 44 | 37.3 | 13.7% |
| 50–64 | 54 | 50.3 | 19.6% | 77 | 69.4 | 20.4% |
| **Education** | | | * | | | *** |
| No education | 19 | 15.2 | 5.3% | 22 | 16.6 | 18.1% |
| Primary | 38 | 34.1 | 11.5% | 36 | 31.5 | 23.2% |
| Secondary | 53 | 43 | 21% | 52 | 50.2 | 8% |
| Higher | 11 | 14.3 | 43% | 8 | 8.3 | 59.4% |
| **BMI status** | | | | | | |
| Underweight | 6 | 3.7 | 0.0% | 9 | 7.8 | 22.7% |
| Normal | 26 | 27.7 | 16.1% | 49 | 35.4 | 17.4% |
| Overweight | 35 | 24.9 | 27.7% | 38 | 47.6 | 16.4% |
| Obese | 54 | 50.4 | 17% | 25 | 15.8 | 22.4% |
| **Current smoking status (last 24hrs)** | | | | | | |
| Don't smoke | 100 | 93 | 19.1% | 80 | 74.9 | 19.7% |
| Do smoke | 21 | 13.6 | 15.8% | 41 | 31.8 | 14.3% |
| **Diabetes status[1]** | | | | | | |
| No Diabetes | 103 | 92.2 | 17.9% | 105 | 97.5 | 17.1% |
| Have Diabetes | 18 | 14.5 | 23.3% | 16 | 9.1 | 29.1% |
| **Couples Level Characteristics** | | | | | | |
| **Residence** | | | | | | |
| Urban | 76 | 74.7 | 19.4% | 76 | 74.7 | 17.5% |
| Rural | 45 | 32 | 16.8% | 45 | 32 | 19.5% |
| **Household wealth** | | | | | | |
| Lower | 45 | 39.2 | 13.5% | 45 | 39.2 | 12.2% |
| Upper | 76 | 67.5 | 21.6% | 76 | 67.5 | 21.5% |
| **Household wealth quintiles** | | | | | | |
| Poorest | 7 | 6.81 | 12.1% | 7 | 6.81 | 31.0% |
| Poorer | 22 | 19.5 | 7.2% | 22 | 19.5 | 9.5% |
| Middle | 16 | 12.8 | 23.9% | 16 | 12.8 | 6.2% |
| Richer | 31 | 28.6 | 14.0% | 31 | 28.6 | 14.4% |
| Richest | 45 | 39.0 | 27.2% | 45 | 39.0 | 26.7% |
| **Partner's Characteristics** | | | | | | |
| **Partner's Age** | | | | | | |
| 35–49 | 44 | 37.3 | 13.0% | 67 | 56.4 | 15.9% |
| 50–64 | 77 | 69.4 | 21.7% | 54 | 50.3 | 20.5% |

*(Continued)*

**Table 4.** (Continued)

| | Females | | | Male | | |
|---|---|---|---|---|---|---|
| | Unweighted N (121) | Weighted N (106.7) | Weighted Percentage of N with Controlled Hypertension | Unweighted N (121) | Weighted N (106.7) | Weighted Percentage of N with Controlled Hypertension |
| **Partner's Education** | | | | | | |
| No education | 22 | 16.6 | 5.30% | 19 | 15.2 | 14.7% |
| Primary | 36 | 31.5 | 11.5% | 38 | 34.1 | 17% |
| Secondary | 52 | 50.2 | 21% | 53 | 43 | 12.8% |
| Higher | 8 | 8.3 | 43% | 11 | 14.3 | 40% |
| **Partner's BMI status** | | | | | | |
| Underweight | 9 | 7.8 | 0.0% | 6 | 3.7 | 8.7% |
| Normal | 49 | 35.4 | 17.6% | 26 | 27.7 | 27.1% |
| Overweight | 38 | 47.6 | 16.5% | 35 | 24.9 | 12.5% |
| Obese | 25 | 15.8 | 36.8% | 54 | 50.4 | 16.6% |
| **Partner's Current Smoking status (last 24hrs)** | | | | | | |
| Don't smoke | 80 | 74.9 | 19.1% | 100 | 93 | 18.2% |
| Do smoke | 41 | 31.8 | 17.7% | 21 | 13.6 | 17.2% |
| **Partner's Diabetes status[1]** | | | | | | |
| No Diabetes | 105 | 97.5 | 19.1% | 103 | 92.2 | 17.9% |
| Have Diabetes | 16 | 9.1 | 14.2% | 18 | 14.5 | 19.5% |
| **Couples Level Characteristics** | | | | | | |
| **Residence** | | | | | | |
| Urban | 76 | 74.7 | 19.4% | 76 | 74.7 | 17.5% |
| Rural | 45 | 32 | 16.8% | 45 | 32 | 19.5% |
| **Household wealth** | | | | | | |
| Lower | 45 | 39.2 | 13.5% | 45 | 39.2 | 12.2% |
| Upper | 76 | 67.5 | 21.6% | 76 | 67.5 | 21.5% |
| **Total** | | | | | | |
| | 121 | 106.7 | 18.6% | 121 | 106.7 | 18.1% |

***P < 0.01,

**P < 0.05,

*P < 0.1

[1]—An individual was classified as living with diabetes if they had a fasting plasma glucose of 7 mmol/L or above at the time of the survey. Or the individual had a 'normal' fasting plasma glucose, below 7 mmol/L, at the time of the survey and was on medication to control their diabetes.

Older age is a widely recognised risk factor for hypertension, this relationship is largely associated with structural changes within arteries as well as calcification over time [14, 16, 18]. A 2007 systematic review of 25 studies across 10 SSA countries reported that urban residence and older age are the most significant determinants of higher hypertension prevalence [29].

In addition to being an independent risk factor for NCDs, high BMI ($\geq$30 kg/m2) has repeatedly been associated with increased odds of hypertension [15, 19, 27]. Obesity has also been shown to be a risk factor with high spousal concordance [30]. Individuals living with both diabetes and hypertension is another common pattern of comorbidity [21]. Diabetes is, therefore, a significant predictor of hypertension in many studies, including the 2013 DHS in which females with diabetes were more than twice as likely to be hypertensive (OR 2.23 CI 1.40–3.40) than females without diabetes [15, 22]. The shared disease mechanisms and

primary risk factors, such as obesity, mean that both diabetes and hypertension can be viewed to have a causal relationship with the other [21].

Smoking was not a significant risk factor in our study and whilst hypertension and smoking status are risk factors for cardiovascular disease, the influence of smoking on hypertension status is unclear [23]. In contrast to findings, a prospective cohort study of 28,236

**Table 5. Odds of female hypertension control (bivariable and multivariable logistic regression models) (N = 121).**

| Factors considered | Bivariable Models adjusted for male hypertension control [1] | Multivariable Model A (adjusting for Female Factors) | Multivariable Model B (Model A adjusting for couple factors) | Multivariable Model C (Model B adjusting for male factors) | Final Multivariable Model [3] |
|---|---|---|---|---|---|
| **Male partner's Hypertension Control Status (ref: not controlled)** | 3.00 (1.08–8.36)** | 3.40 (1.13–10.19)** | 3.47 (1.14–10.60)** | 4.42 (1.32–14.76)** | 3.69 (1.23–11.12)** |
| **Female's Characteristics** | | | | | |
| 50–64 years (ref: 35–49) | 0.76 (0.30–1.89) | 0.75 (0.29–1.89) | 0.70 (0.26–1.89) | 0.64 (0.19–2.17) | |
| At least primary education [4] (ref: No education) | 5.67 (0.71–45.56) | 4.23 (0.51–35.09) | 4.78 (0.56–40.95) | 5.18 (0.58–46.55) | |
| Overweight/ Obese [5] (ref: Underweight / Normal) | 3.70 (0.98–13.89)* | 3.08 (0.81–11.81)** | 2.96 (0.76–11.49) | 2.31 (0.55–9.68) | 2.58 (0.65–10.31) |
| Current smoker (last 24hrs) (ref: Doesn't smoke) | 0.86 (0.25–2.89) | 0.86 (0.23–3.18) | 0.86 (0.23–3.19) | 0.81 (0.19–3.40) | |
| Living with Diabetes [2] (ref: Not living with Diabetes) | 2.13 (0.69–6.57) | 1.66 (0.51–5.40) | 1.70 (0.52–5.57) | 1.34 (0.34–5.31) | |
| **Couples Level Characteristics** | | | | | |
| Rural Residence (ref: Urban) | 0.93 (0.36–2.36) | | 1.44 (0.50–4.15) | 1.16 (0.37–3.64) | |
| Greater household wealth [6] (ref: poorest) | 1.92 (0.21–17.67) | | 1.47 (0.11–19.07) | 1.66 (0.11–24.98) | |
| **Male Partner's Characteristics** | | | | | |
| 50–64 (ref: 35–49) | 1.19 (0.46–3.09) | | | 1.74 (0.48–6.34) | |
| Higher Education [7] (ref: No education / Primary/ Secondary) | 1.01 (0.18–5.69) | | | 0.61 (0.09–4.06) | |
| Obese [8] (ref: Underweight/ Normal weight/ Overweight) | 3.94 (1.44–10.80)*** | | | 2.91 (0.84–10.04)* | 3.06 (1.07–8.76)** |
| Current smokes (last 24hrs) (ref: Not a current smoker) | 0.95 (0.36–2.47) | | | 1.39 (0.43–4.42) | |
| Living with Diabetes [2] (ref: Not living with Diabetes) | 0.77 (0.19–3.05) | | | 0.41 (0.09–1.97) | |

***P < 0.01,

**P < 0.05,

*P < 0.1

[1]—Bivariable models—adjusting for partner hypertension control (key association of interest) + relevant factor for that row

[2]—An individual was classified as living with diabetes (Y/N) if they had a fasting plasma glucose of 7 mmol/L or above at the time of the survey. Or the individual had a 'normal' fasting plasma glucose, below 7 mmol/L, at the time of the survey and was on medication to control their diabetes.

[3]—Final model—adjusting for partner hypertension status (key association of interest) + variables that contributed significantly at the 10% level to the models, using a likelihood ratio test

[4]—At least primary' created by combining primary, secondary and higher education categories

[5]—'Overweight/ Obese' created by combining overweight and obese BMI categories (for the reference underweight and normal weight BMI categories were combined)

[6]—'Greater household wealth' created by combining the poorer, middle, richer and richest DHS wealth quintiles

[7]—Male partner 'Higher Education' created by combining no education, primary, secondary education categories

[8]—Male partner 'Obese' created by combining underweight, normal weight and overweight BMI categories

**Table 6. Odds of male hypertension control (bivariable and multivariable logistic regression models) (N = 121).**

| Factors considered | Bivariable Models adjusted for female hypertension control[1] | Multivariable Model A (adjusting for Male Factors) | Multivariable Model B (Model A adjusting for couple factors) | Multivariable Model C (Model B adjusting for Female factors) | Final Multivariable Model [3] |
|---|---|---|---|---|---|
| **Female partner's Hypertensive Control Status (ref: not controlled)** | **3.00** (1.08–8.36)** | **3.57** (1.18–10.80)** | **3.66** (1.20–11.19)** | **4.03** (1.26–12.88)** | **3.00** (1.08–8.36)** |
| **Male's Characteristics** | | | | | |
| 50–64 years (ref: 35–49) | 1.48 (0.52–4.21) | 1.33 (0.45–3.90) | 1.32 (0.43–4.02) | 1.27 (0.36–4.50) | |
| Higher Education[4] (ref: No education / Primary/ Secondary) | 3.16 (0.66–15.07) | 3.86 (0.65–23.01) | 3.95 (0.66–23.74) | 4.00 (0.61–26.25) | |
| Obese[5] (ref: Underweight / Normal weight / Overweight) | 0.62 (0.17–2.22) | 0.44 (0.11–1.79) | 0.46 (0.11–1.92) | 0.35 (0.07–1.71) | |
| Current smoker (last 24hrs) (ref: Doesn't smoke) | 0.75 (0.26–2.15) | 0.77 (0.25–2.30) | 0.74 (0.24–2.25) | 0.61 (0.18–2.02) | |
| Living with Diabetes [2] (ref: Not living with Diabetes) | 1.82 (0.51–6.53) | 1.75 (0.46–6.65) | 1.88 (0.48–7.34) | 1.89 (0.44–8.05) | |
| **Couples Level Characteristics** | | | | | |
| Rural Residence (ref: Urban) | 1.06 (0.40–2.86) | | 0.94 (0.31–2.84) | 0.84 (0.26–2.71) | |
| Greater household wealth[6] (ref: poorest) | 1.44 (0.08–2.56) | | 0.39 (0.06–2.55) | 0.30 (0.04–2.20) | |
| **Female Partner's Characteristics** | | | | | |
| 50–64 (ref: 35–49) | 1.23 (0.47–3.22) | | | 1.02 (0.30–3.46) | |
| At least primary education[7] (ref: No education) | 0.89 (0.22–3.51) | | | 0.67 (0.14–3.12) | |
| Obese[8] (ref: Underweight / Normal weight/ Overweight) | 1.12 (0.43–2.93) | | | 1.40 (0.45–4.39) | |
| Current smokes (last 24hrs) (ref: Not a current smoker) | 1.19 (0.35–4.07) | | | 1.56 (0.41–5.96) | |
| Living with Diabetes[2] (ref: Not living with Diabetes) | 1.22 (0.34–4.33) | | | 1.62 (0.37–7.03) | |

***P < 0.01,

**P < 0.05,

*P < 0.1

[1]—Bivariable models—adjusting for partner hypertension control (key association of interest) + relevant factor for that row

[2]—An individual was classified as living with diabetes (Y/N) if they had a fasting plasma glucose of 7 mmol/L or above at the time of the survey. Or the individual had a 'normal' fasting plasma glucose, below 7 mmol/L, at the time of the survey and was on medication to control their diabetes.

[3]—Final multivariable model—There were no variables that contributed significantly at the 10% level to the models, using a likelihood ratio test

[4]—'Higher Education' created by combining no education, primary, secondary education categories

[5]—'Obese' created by combining underweight, normal weight and overweight BMI categories

[6]—'Greater household wealth' created by combining the poorer, middle, richer and richest DHS wealth quintiles

[7]—Female partner 'At least primary' created by combining primary, secondary and higher education categories

[8]—Female partner 'Obese' created by combining underweight, normal weight and overweight BMI categories

American women, found that the risk of hypertension increases in women who smoke more than 15 cigarettes a day (aOR 1.11 (CI 1.03–1.21)) compared to those who have never smoked [24].

## Spousal concordance in hypertension control

Yuyun et al. reviewed articles covering the prevalence of cardiovascular diseases (CVD) in SSA from January 1990 to March 2019 and reported that over 60% of hypertensive adults (>18 years old) were unaware of their condition [31]. The low rates of CVD awareness in SSA were attributed to insufficient health care infrastructure and lack of resource allocation towards NCDs. Low rates of awareness are mirrored in the Namibia DHS final report with 49% of hypertensive females and 61% of hypertensive males being unaware that they had elevated blood pressure [14].

Yuyun et al. also found that in SSA only 10–20% of individuals diagnosed with hypertension had their BP controlled; low rates of control were also found in this Namibian study [31]. Our study found a significant association between hypertension control status between partners. Hypertensive females whose male partners had controlled hypertension, were 3.69 times more likely to have controlled BP compared to hypertensive females whose partners had uncontrolled hypertension, after adjustment for female BMI and diabetes, OR 3.69 (CI 1.23–11.12) (Table 5). In a cohort study by McAdams et al. in the United States, from 1986–2011, which measured married participants' (aged 45–64 years) blood pressure in four visits and suggested a positive association between individual and partner hypertension control although this was not statistically significant, before or after adjustment for both individual and partner risk factors (age, race, BMI, smoking status, and sodium intake), OR 1.22 (CI 0.95–1.57) vs aOR 1.21 (CI 0.93–1.56) respectively, and for both male and female hypertension control outcomes [8]. The difference in association found by McAdams et al. may be due to their longitudinal study design, larger sample size of 4500 pairs and adjustment for more risk factors, such as salt intake [8]. Explanations for spousal concordance for hypertension control may include partners having equal access to healthcare services; or partners sharing health-seeking behaviours, such as adherence to antihypertensives [7].

## Hypertension interventions in Namibia

The Pan—African Society of Cardiology (PASCAR) 'Roadmap to decrease the burden of hypertension in Africa' found that the majority of African countries did not have an active hypertension policy programme in 2017, including Namibia [32]. Ten actions to be undertaken by African ministries of health were published, the first being 'All NCD national programmes should additionally contain a plan for the detection of hypertension'.

PASCAR recognised that achieving greater levels of hypertension control as the 'highest area of priority' in an effort to minimise the currently rising rates of heart disease and stroke across Africa [32]. The significant roadblocks standing in the way, identified within the PASCAR roadmap, fall under three subgroups: government and health system-related, health care professional-related and patient-related [32]. The roadmap also acknowledges that there is a domino effect in which the lack of hypertension policy (government—related); poor universal health coverage (health system-related) and the low doctor to patient ratio (health professional-related) all result in a lack of hypertension education, awareness and adherence (patient related) and continued increasing rates of hypertension.

In terms of the prevention and control of hypertension in Namibia, there is minimal reference to hypertension screening and management within the Ministry of Health's (MoH) NCD plan for 2017/18–2021/22 [33]. Four behavioural risk factors for NCDs were recognised within

the plan: 'use of tobacco products, harmful use of alcohol, physical inactivity and unhealthy diets' [33]. Nine targets were set including '*Halt the rise in obesity and Diabetes Mellitus by 2022*' and '*A 15% relative reduction in prevalence of raised blood pressure and/or contain the prevalence of raised blood pressure by 2022; and a 25% relative reduction by 2025*'. There is a need for specific hypertension policies in Namibia, given the high prevalence of hypertension among males and females within our sample (51% and 45% respectively), consistent with the population estimates from the 2013 DHS report [14]. The Namibian National Health Policy Framework recognises the rising levels of NCDs and lists action points which include surveillance of NCD risk factors, institutionalization of NCD screening and 'strengthening health promotion through behavioural change communication, including community dialogue' [34].

Past studies of spousal hypertension have highlighted potential cost and efficiency benefits to spousal screening for hypertension, which could be evaluated for Namibia [5, 8]. Knowledge of spousal concordance for hypertension could act as an important guide within such interventions, through inviting the partners of all hypertensive individuals for screening or targeting nurse-led hypertension education at couples [35].

Existing nurse led hypertension interventions in SSA have been designed to overcome obstacles, such as low levels of hypertension awareness and low doctor to patient ratios. A retrospective study of 1051 hypertensive adults (aged over 35 years) in rural Kenya who all enrolled in a nurse led hypertension management program found that there was significant SBP reduction in participants from baseline to 3 months of—4.95 mmHg/month (95% CI: 6.55 to—3.35) [35]. A randomised control trial conducted among 757 participants in Ghana, found that receiving nurse led management in addition to health insurance coverage was associated with the greatest reduction in SBP [36]. Such nurse led interventions improve rates of hypertension through patient education and more regular follow ups. They offer cost-effective solutions to overcome the low doctor to patient ratios in SSA and their potential impact could be enhanced by incorporating partner involvement.

Additionally in couples where only one partner is hypertensive, spousal support could be utilised in couples-focused interventions to support adherence to hypertension treatment and lifestyle change. By applying the interdependence and communal coping theories, a couples-focused behavioural change intervention could promote a transition towards couples viewing hypertension as a shared problem [11, 12, 34]. This would encourage couples to have a relationship-focused motivation to control hypertension and reduce the risk factors for the second partner. A cross sectional study of 435 hospitalised patients in China found a strong positive association between social support and hypertensive treatment adherence, OR 0.75 (95% CI: 0.68–0.83) [37]. The study found that this support was mainly provided by an individual's nuclear family, i.e., spouses, partners and children. Arabshahi et al. conducted a cross-sectional study in Iran and found a significant relationship between total social support score from the spouse and decreased SBP—0.151 (P = 0.01) and DBP—0.179 (P = 0.003) and recommended that future hypertension interventions account for the value of good spousal social support [38].

## Strengths and limitations

The 2013 DHS survey round is the first and only national survey in Namibia, to date, to include biomarker data collection, making its contribution to understanding the distribution and patterns of hypertension in this setting all the more important [14].

Whilst the DHS aims to be representative of the population overall; the analysis sample is not representative of all Namibian couples. All couples in our analyses were aged between 35–64 years due to the age-related eligibility for the biomarker survey, which means the findings

may not be generalisable to other age groups, as the risk of hypertension increases independently with age [16, 18]. Data analyses were restricted by the number of couples who met all the inclusion criteria, resulting in limited power for some analyses. Sample size was reduced further for the hypertension control question as only couples with two hypertensive partners were eligible, limiting our ability to explore multivariable models of hypertension control. The use of binary variables, on occasion, did not make use of all the information available but was necessary for modelling given the sample number constraints.

Our analyses used a cross-sectional design and therefore gives a snapshot of hypertension prevalence. As the onset of hypertension and length of marriage is unknown this is a study of concordance rather than of a causal relationship between marriage and hypertension. Partner concordance in this study is likely to mask heterogeneity in multiple dimensions of partnerships. Other studies have suggested that further marriage variables, such as marital satisfaction and spousal contact, may be better predictors for hypertension than marital status alone [38–40], however these variables were not available in the DHS dataset.

As the DHS took three BP measurements on the same day (rather than on two different days required in the WHO definition); both 'Hypertensive' and 'Controlled' are not clinical definitions within this study but offer an indication of the proportion of individuals within the sample with hypertension for the first research question and with controlled hypertension for the secondary research question.

Nonetheless, our analyses are the first to explore spousal concordance in hypertension in SSA; therefore, the findings from this exploratory study contribute knowledge of spousal concordance in hypertension in Namibia and suggest further research of this kind in SSA would be beneficial to MoH planning.

## Conclusion

Having a hypertensive partner was positively associated with increased odds of hypertension in individuals, among married and cohabiting Namibian adults aged 35–64 years. Similarly, partner's hypertension control was significantly associated with greater odds of individual hypertension control. High rates of hypertension and low rates of control are a growing concern in SSA, and hypertension control has been recognised as a top priority in order to reduce the number of heart attacks and strokes across Africa [32]. Current Namibian policy has listed actions to reduce four behavioural risk factors for NCDs [33]. Despite no reference to hypertension specific interventions, the actions listed within the plan address significant hypertension risk factors seen in this study such as diabetes, high BMI and lack of education [33]. Government health policies and the development of behaviour change interventions in SSA are needed to increase rates of hypertension awareness and control. Couples—focused interventions such as routine screening of the partners of hypertensive individuals and utilising spousal support in hypertensive treatment adherence, could be potential cost effective and efficient strategies.

## Author Contributions

**Formal analysis:** Alice Rose Weare, Zhixin Feng, Nuala McGrath.

**Investigation:** Alice Rose Weare, Zhixin Feng, Nuala McGrath.

**Methodology:** Alice Rose Weare, Zhixin Feng, Nuala McGrath.

**Project administration:** Alice Rose Weare.

**Supervision:** Zhixin Feng, Nuala McGrath.

**Writing – original draft:** Alice Rose Weare.

**Writing – review & editing:** Zhixin Feng, Nuala McGrath.

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
