## [Decision Letter · Decision Letter 0]

18 Nov 2022

PONE-D-22-17747The Prevalence of Hypertension and Hypertension Control Among Married Namibian Couples PLOS ONE

Dear Dr. Weare,

Thank you for submitting your manuscript to PLOS ONE. After careful consideration, we feel that it has merit but does not fully meet PLOS ONE’s publication criteria as it currently stands. Therefore, we invite you to submit a revised version of the manuscript that addresses the points raised during the review process.

We look forward to receiving your revised manuscript.

Kind regards,

Melkamu Merid Mengesha, MPH

Academic Editor

PLOS ONE

Journal Requirements

Additional Editor Comments:

The reviewers have raised important concerns on your submission that need to be addressed. The following points were emphasized: weak data interpretation and that conclusions are not based on data; editorial and citation related issues; clarity on variable measurement and operational definition; survey weight; lack of adequate description about the study setting and population including inclusion and exclusion criteria; data quality; and the need for a substantial improvement of the discussion as it has been a shallow presentation. The authors should provide a point-by-point response to these and other comments of the reviewers in their revised submission.

Reviewers' comments:

Reviewer's Responses to Questions

**Comments to the Author**

1. Is the manuscript technically sound, and do the data support the conclusions?

Reviewer #1: Partly

Reviewer #2: Yes

2. Has the statistical analysis been performed appropriately and rigorously? 

Reviewer #1: I Don't Know

Reviewer #2: Yes

3. Have the authors made all data underlying the findings in their manuscript fully available?

Reviewer #1: Yes

Reviewer #2: Yes

4. Is the manuscript presented in an intelligible fashion and written in standard English?

Reviewer #1: Yes

Reviewer #2: Yes

5. Review Comments to the Author

Reviewer #1: Weare et al. presented the study to explore the spousal concordance for hypertension (HP) and hypertension control in Namibia. The data from Namibia Demographic and Health Survey were analyzed to investigate the relationship between spouse, HP as well as disease control. They identified that, in Namibian adults, the partnership is correlated with HP and its control. The method is straightforward and the figures are easy to follow. This paper addressed a significant clinical problem, and the results are original with novelty in the target population. My major concern in this study is that the data interpretation is weak and the conclusion could not be fully supported based on the current context.

As my expertise is hypertension, I will mostly comment on that part of the methods and data.

1. In methods for the built variable models: although bivariable models and multivariable models have been widely used, the models are more than what is included in the methods. Please explain the rationale for using each model (especially for the multivariable model), and provide citations of similar studies using the same model before.

2. In lines 182-183 “with the exception of partner age”: from Table 1 partner characteristics, BMI status also showed a difference here, while authors did not cover the exception. Please clarify.

3. In lines 183-184: the authors argued the HP prevalence in partner status, while no data was presented in the table. Please explain how these prevalence data are generated.

4. In lines 199-20: Same as point 4.

5. Please explain what is the gray box in the final columns of the tables.

6. In Table 4, The Pearson test showed the significance of BMI status here. Please elaborate on it in the context.

7. In line 257, the authors argued the “3.67 times”. Please explain this number was generated.

8. In line 260, please clarify how the consistency between the US study (non-significant) and your study results (assume significantly).

9. Please double-check for errors including typos, extra marks, and grammar

Reviewer #2: Thank you the editor in chief, for providing this golden opportunity to me to review the interesting manuscript. This paper used very strong multilevel model to assess the prevalence, and control of HTN in Namibian couples. The researcher also identified those factors which had association with the prevalence and control of HTN in couples. There are very interesting finds which can play a great role in alleviating the increasing burden of NCD. However, to assure its contribution for readers, still it needs a great work. For this matter I tried to put my concerns here below headed as minor and major comments. I hope the author will cover all points and make the manuscript sounder. Looking for the modified document!!

Minor Comments:

• Line 35-41: In abstract, the result section does not include all relevant findings in line with the topic of the study.

• Line 88-96: Citation has to be put.

• Line 99-102: Why the research question was mixed with the Methods? It has to be separated!

• Line 169 (Fig 1): Mention it in the method section.

• Line 171: What type of weighing was used? Why?

• Line 171-184: The citation of table missed.

• Line 171: The appropriate heading needs to be given for the first objective which was “The prevalence of HTN in couples” and you need to compare both groups too with respect to the outcome variable.

• Line 171-177: You mentioned as the males and females experienced the variability of HTN level across the categories of individual characteristics. But, there is no any comparison statistics in table 1 that put in the narration. It would be better to put the actual P-values in the table.

• Table 1: How the variables like age, BMI, wealth index were measured? The unit? Additionally mention them in method section. Better to make the table topic more self-explanatory! When? Where?

• Line 201: You defined uncontrolled HTN as those who were either unaware of their hypertension status or those who were aware but not controlled. Here both partners were unaware of their hypertension in 37.7% of couples. What about those who were aware but not controlled? And also in line 203 both partners were in control of their hypertension in only 8.31% of hypertensive couples. Which means those with uncontrolled HTN would be about 91.7%? What about 37.7%?

• Line 189: Individuals? Male? Female?

• Line 229: The association was simply marginal, you discusses as if they had association. What??

• Line 239: In contrast? Two similar issues are compared. Why you mentioned as a contrast?

• Line 241: Regarding residency, rural or urban category was obtained as a factor? Correct it.

• Line 257: 3.67?

Major comments:

• The author operationalized the “HTN control” in line 118-119. However, how those who were aware but not controlled were identified in NDHS survey? It has to be mentioned in a clear way. Generally, the way how an author categorized either controlled or uncontrolled HTN is not clear. The survey was not facility based and it was a snapshot. So, how confident is the author to measure and report the individuals’ HTN control status.

• Even though secondary data was used, the author has to explain the following points in method part deeply: The study setting? The setting characteristics?, which population data set was used in this analysis?, how many of them fulfilled the inclusion criteria, how many of them were removed/dropped? (492 couples for HTN prevalence Vs 121 for HTN control), what study design was applied ?, how the study subjects were recruited? (all stages of sampling need to be explained in detail), what type of weighting was applied? And why?, how the data quality was assured in NDHS?, how missed variable were managed?, why a Multilevel LR model was applied?, how much was the cluster correlation level (within-cluster correlation)?, how you measured?, at what level of intra-cluster correlation the multilevel analysis is recommended?, what individual and community level factors were considered, how you assessed your model fitness? (The steps of model building have to be explained clearly). Generally, the method section is poor in mentioning above issues. Hence, the authors must incorporate these points seriously.

• Discussion need to be sequenced based on the order of objectives of this study.

• The discussion is shallow especially for the factors obtained for the HTN. The possible explanations for each variable were not discussed deeply. So, re-write it.

• In discussion, you only tried to discuss those variables for females HTN. What about those factors for males’ HTN?

• In conclusion section, the authors have to make sure that all mentioned recommendations considered all identified factors.

• Did the researcher answer the research question? Which was the couple’s concordance in HTN prevalence and HTN control? I do not think so. How can we measure either the concordance exists or not? For me the researcher simply assessed the prevalence of HTN and associated factors for both sexes (males and females), and even though I do not agreed with the measurement of the variable HTN control status, the control status and factors were also assessed. I need clear explanation on these issue, and the author has to make the findings of this study more easily readable for the readers of this document.

• The topics for tables and figures have to be made self-explanatory and the fig. topic needs to be put at the appropriate place, preferably at the bottom of figures.

• For web page references, the URL, Access date and citation dates have to be incorporated.

• English editing is also highly needed.

6. PLOS authors have the option to publish the peer review history of their article (what does this mean?). If published, this will include your full peer review and any attached files.

Reviewer #1: No

Reviewer #2: **Yes: **Mathewos Alemu Gebremichael

---

## [Author Response · Author response to Decision Letter 0]

18 Mar 2023

Dear PLOS ONE editor and reviewers,

Thank you for taking the time to read and review our submission ‘The Prevalence of Hypertension and Hypertension Control Among Married Namibian Couples’. We have addressed the reviewers concerns with a point-by-point response below and the revised manuscript includes tracked changes. 

Journal Requirements

Authors’ response: We have ensured the manuscript meets PLOS ONE's style requirements, matching the examples given, in particular making changes to our citation format and subheading format.

Authors’ response: We have now provided this additional information under the ethics section in our methods, page 7. ‘All participants gave written consent prior to taking part in the DHS questionnaires and having their blood pressure measured.(14) All data were fully anonymized by the DHS program before we accessed them for our study. (14)’

3. In your Data Availability statement, you have not specified where the minimal data set underlying the results described in your manuscript can be found. PLOS defines a study's minimal data set as the underlying data used to reach the conclusions drawn in the manuscript and any additional data required to replicate the reported study findings in their entirety. All PLOS journals require that the minimal data set be made fully available. For more information about our data policy, please 

see http://journals.plos.org/plosone/s/data-availability. "Upon re-submitting your revised manuscript, please upload your study’s minimal underlying data set as either Supporting Information files or to a stable, public repository and include the relevant URLs, DOIs, or accession numbers within your revised cover letter. For a list of acceptable repositories, please

see http://journals.plos.org/plosone/s/data-availability#loc-recommended-repositories. Any potentially identifying patient information must be fully anonymized.

Authors’ response: Our authorisation to download and use the Namibian Survey data from the DHS Program was granted on the 29th April 2020. Reference number:142031. 

Our updated Data Availability statement…

This was a retrospective study using third party data from the 2013 Namibian Demographic and Health Survey. DHS data are publicly available through the (https://dhsprogram.com/methodology/survey/survey-display-363.cfm ). 

Authors’ response: Our ethics statement is now included within the Methods, on page 4.

‘The 2013 Namibian DHS questionnaires and procedures were reviewed and approved by the ICF Institutional Review Board and the Ministry of Health and Social Services Biomedical Research Committee. All participants gave written consent prior to taking part in the DHS questionnaires and having their blood pressure measured.(14) All data were fully anonymized by the DHS program before we accessed them for our study. (14) Ethics approval for our study was granted by the University of Southampton Ethics and Research Governance Online (ERGO) committee.’

Additional Editor Comments:

The reviewers have raised important concerns on your submission that need to be addressed. The following points were emphasized: weak data interpretation and that conclusions are not based on data; editorial and citation related issues; clarity on variable measurement and operational definition; survey weight; lack of adequate description about the study setting and population including inclusion and exclusion criteria; data quality; and the need for a substantial improvement of the discussion as it has been a shallow presentation. The authors should provide a point-by-point response to these and other comments of the reviewers in their revised submission.

Authors’ response: Thank you for the additional editor comments, we have set out to address each of these concerns within our revised manuscript. 

5. Review Comments to the Author

Reviewer #1: Weare et al. presented the study to explore the spousal concordance for hypertension (HP) and hypertension control in Namibia. The data from Namibia Demographic and Health Survey were analysed to investigate the relationship between spouse, HP as well as disease control. They identified that, in Namibian adults, the partnership is correlated with HP and its control. The method is straightforward and the figures are easy to follow. This paper addressed a significant clinical problem, and the results are original with novelty in the target population. My major concern in this study is that the data interpretation is weak and the conclusion could not be fully supported based on the current context.

Authors’ response: We are really pleased that you felt our focus on couples has novelty and that our paper investigates a significant clinical problem. Thank you for all your comments, we believe changes made in response to your comments have strengthened our paper.

1. In methods for the built variable models: although bivariable models and multivariable models have been widely used, the models are more than what is included in the methods. Please explain the rationale for using each model (especially for the multivariable model), and provide citations of similar studies using the same model before.

Authors’ response: We have now included citations to past hypertension concordance studies that we used when deciding on the data analysis for our study (page 6). We have also expanded on the rationale for the models used.

2. In lines 182-183 “with the exception of partner age”: from Table 1 partner characteristics, BMI status also showed a difference here, while authors did not cover the exception. Please clarify.

Authors’ response: We have edited the text to also include partner BMI status as this was another exception (Page 11 – ‘In general, male hypertension prevalence differed significantly across levels of all the female partner characteristics considered, while female hypertension prevalence remained similar across levels of the male partner characteristics except for partner age (p= 0.04) and partner BMI (p=0.05).’)

3. In lines 183-184: the authors argued the HP prevalence in partner status, while no data was presented in the table. Please explain how these prevalence data are generated.

Authors’ response: Thank you for your comment. We have edited the text and now refer to the data presented in the table. 

4. In lines 199-20: Same as point 4.

Authors’ response: This text has been moved to ‘Spousal concordance in hypertension control’ 

page 16 and refers to percentages within Table 4.

5. Please explain what is the gray box in the final columns of the tables.

Authors’ response: The grey boxes were used to blank out empty boxes for variables not included in the final multivariable model, these grey boxes have been removed to make the tables clearer.

6. In Table 4, The Pearson test showed the significance of BMI status here. Please elaborate on it in the context.

Authors’ response: BMI was not significant for hypertension control and the table has been corrected accordingly, page 17.

7. In line 257, the authors argued the “3.67 times”. Please explain this number was generated.

Authors’ response: This was a typo in text and has now been corrected to 3.69, the OR has also been included at the end of the sentence. Page 23 - the sentence has been corrected to ‘Hypertensive females whose male partners had controlled hypertension, were 3.69 times more likely to have controlled BP compared to hypertensive females whose partners had uncontrolled hypertension, after adjustment for female BMI and diabetes, OR 3.69 (CI 1.23 - 11.12) (Table 5).’

8. In line 260, please clarify how the consistency between the US study (non-significant) and your study results (assume significantly).

Authors’ response: We have added text to clarify the differences between our study and the US study. Page 23- ‘The difference in association found by McAdams et al. may be due to their longitudinal study design, larger sample size of 4500 pairs and adjustment for more risk factors, such as salt intake.(8)

9. Please double-check for errors including typos, extra marks, and grammar

Authors’ response: We have revised text in places for clarity and corrected errors.

Reviewer #2: Thank you the editor in chief, for providing this golden opportunity to me to review the interesting manuscript. This paper used very strong multilevel model to assess the prevalence, and control of HTN in Namibian couples. The researcher also identified those factors which had association with the prevalence and control of HTN in couples. There are very interesting finds which can play a great role in alleviating the increasing burden of NCD. However, to assure its contribution for readers, still it needs a great work. For this matter I tried to put my concerns here below headed as minor and major comments. I hope the author will cover all points and make the manuscript sounder. Looking for the modified document!!

Authors’ response: Thank you for taking the time to review are manuscript, we are delighted that you found it interesting and believe the findings can play a role in alleviating the increasing burden of NCDs. Thank you for your comments, they have been very helpful in revising and improving the paper.

Minor Comments:

• Line 35-41: In abstract, the result section does not include all relevant findings in line with the topic of the study.

Authors’ response: Thank you for highlighting this, we have adjusted the abstract, page 2, working with the word limit, to include the significant factors for HTN within the results section. The results section now reads ‘Results : The unadjusted odds ratio of 1.57 (CI 1.10 – 2.24) for hypertension among individuals (both sexes) whose partner had hypertension compared to those whose partner did not have hypertension, was attenuated to aOR 1.35 (CI 0.91 – 2.00) for females (after adjustment for age, BMI, diabetes, residence, individual and partner education) and aOR 1.42 (CI 0.98 – 2.07) for males (after adjustment for age and BMI). Females and males were significantly more likely to be in control of their hypertension if their partner also had controlled hypertension, aOR 3.69 (CI 1.23 - 11.12) and aOR 3.00 (CI 1.07 - 8.36) respectively.’ 

• Line 88-96: Citation has to be put.

Authors’ response: Thank you, citation to the DHS final report has now been included here, page 6 under the Questionnaires heading. ‘There were three DHS questionnaires administered: household, men’s and women’s.(14)’

• Line 99-102: Why the research question was mixed with the Methods? It has to be separated!

Authors’ response: Thank you for this comment, the research questions have now been separated from the methods section, on page 5.

• Line 169 (Fig 1): Mention it in the method section.

Authors’ response: Thank you for your comment, Figure 1 is now mentioned in the methods, page 6.

• Line 171: What type of weighing was used? Why?

Authors’ response: Thank you for your comment, subheadings have now been added to the methods section and the weighting used is explained under ‘Sample Design and Weight’, pages 5/6.

• Line 171-184: The citation of table missed. 

Authors’ response: Thank you for this comment, Table 1 is now cited and all the text in the paragraph refers to data within the table, page 11. 

• Line 171: The appropriate heading needs to be given for the first objective which was “The prevalence of hypertension in couples” and you need to compare both groups too with respect to the outcome variable.

Authors’ response: Thank you for this comment, this section of text refers to data in Table 1 - Proportion of Females and Males with a Hypertensive Status based on Individual and Partner’s Characteristics, now under the subheading ‘Hypertension prevalence based on Individual and 

Partner’s Characteristics’, on page 11.

• Line 171-177: You mentioned as the males and females experienced the variability of hypertension level across the categories of individual characteristics. But, there is no any comparison statistics in table 1 that put in the narration. It would be better to put the actual P-values in the table.

Authors’ response: Thank you for this suggestion, we have included p-values in the table, using Pearson Chi – Square - ***P < 0.01, **P < 0.05, *P < 0.1 (in the table footnotes), this tested the association between individual and partner factors and hypertension status for both males and females. We didn’t use comparison statistics to compare male and female variability of hypertension level. Actual p-value are now given within the text whenever they are discussed on page 11.

• Table 1: How the variables like age, BMI, wealth index were measured? The unit? Additionally mention them in method section. Better to make the table topic more self-explanatory! When? Where?

Authors’ response: Thank you for this comment, we have now expanded in more detail how each variable was measured by the DHS, in the methods under ‘Confounding (Hypertension Risk Factor) Variables’ pages 8/9. This section now says…

‘Confounding (Hypertension Risk Factor) Variables

Key hypertension risk factors identified from the literature(15) and available in the dataset were considered in the model for each research question: (age(16, 18), obesity(19), education(20), diabetes status(21, 22), current smoking status (23-25)) for each individual partner and at the couple’s level: household wealth (6, 20, 26) and urban vs rural residence(27). Age was considered in the model as a binary indicator representing 35-49 years vs 50-64 years, as was residence (urban vs rural). (14) Height (m) and weight (kg) of participants were used to calculate their body mass index (BMI) (kg/m2) and then grouped into the WHO categories of Underweight (BMI<18.5), Normal (18.5-24.9), Overweight (25-29.9) and Obese (>30). (14) For smoking status, current smoking status was considered in models as a binary indicator (Yes vs No). Using the DHS definition, an individual was classified as having diabetes if he/she had a fasting plasma glucose of >7 mmol/L or was currently taking diabetes medication, diabetes status was grouped into ‘No Diabetes’ and ‘Have Diabetes’. (14) 

Education was defined by the individual’s highest level of education attainment at the time of survey and considered in the models as a categorical variable using dummy indicators for 'No education’, ‘Primary’, ‘Secondary” and ‘More than secondary’. (14) Wealth was categorised by the DHS wealth quintile calculations of wealth factors including household assets, into ‘Poorest’, ‘Poorer’, ‘Middle’, ‘Richer’ and ‘Richest’. (14) Adjustment for further known hypertension risk factors such as physical inactivity (PA)(15), alcohol consumption(15) and salt intake(15, 28) were beyond the scope of the study as these data were not collected in this survey (PA and salt intake) or only collected for a subset of the analysis sample (alcohol consumption).(14)’

• Line 201: You defined uncontrolled hypertension as those who were either unaware of their hypertension status or those who were aware but not controlled. Here both partners were unaware of their hypertension in 37.7% of couples. What about those who were aware but not controlled? And also in line 203 both partners were in control of their hypertension in only 8.31% of hypertensive couples. Which means those with uncontrolled hypertension would be about 91.7%? What about 37.7%?

Authors’ response: Thank you for this comment. We have clarified the definition of ‘Controlled’ and ‘Uncontrolled’ within our methods, under ‘Outcome variables’ on page 8. In our results we have focused the text to the results of spousal concordance for hypertension control, to avoid confusion with hypertension awareness. The low rates of hypertension awareness in Namibia are now raised in the discussion section, with reference to the prevalence of hypertension awareness reported in the DHS final report. (These changes are detailed below under the major comments).

• Line 189: Individuals? Male? Female?

Authors’ response: Thank you, we have removed ‘individuals’ and used ‘Males and females’ to avoid confusion, page 13 - ‘Both males and females were significantly more likely to have hypertension if their partner was also hypertensive, OR 1.57 (CI 1.10 – 2.24), p= 0.01 (bivariable models in Table 2 and 3).’

• Line 229: The association was simply marginal, you discusses as if they had association. What??

Authors’ response: We have reworded this text to discuss the statistical significance of the association found, first paragraph of page 21. ‘In our analyses, partner hypertension was significantly associated with individual hypertension in unadjusted models (OR 1.57 (CI 1.10 – 2.24), Table 2 and 3) and the estimate of this association was only slightly attenuated in adjusted models, however it was no longer statistically significant (female aOR 1.35 (CI 0.91 – 2.00), in Table 2 and male aOR 1.42 (0.98 – 2.07), in Table 3).’

• Line 239: In contrast? Two similar issues are compared. Why you mentioned as a contrast?

Authors’ response: Thank you, we have removed ‘in contrast’ and changed the text to avoid confusion, in paragraph 2 page 21.

• Line 241: Regarding residency, rural or urban category was obtained as a factor? Correct it.

Authors’ response: Thank you for this comment, residence was used a hypertension risk factor and the female model found urban residence to be significantly associated with increased odds of hypertension, page 21.

• Line 257: 3.67?

Authors’ response: Thank you for highlighting this, this was a typo in text and has now been corrected, the OR has also been included at the end of the sentence, page 23.

Major comments:

• The author operationalized the “HTN control” in line 118-119. However, how those who were aware but not controlled were identified in NDHS survey? It has to be mentioned in a clear way. Generally, the way how an author categorized either controlled or uncontrolled HTN is not clear. The survey was not facility based and it was a snapshot. So, how confident is the author to measure and report the individuals’ HTN control status.

Authors’ response: Thank you for this comment. We have clarified that we follow the DHS definition of controlled and uncontrolled in the methods (page 8) and discuss the limitations of this approach and potential for some misclassification in the limitations (page 26). ‘Similarly, following DHS operationalisation of hypertension control using average BP measurements and antihypertensive medication self-report(14), a binary variable was created to categorise each hypertensive individual as having their hypertension ‘Controlled’ or ‘Uncontrolled’. Individuals were asked ‘Have you ever been told by a doctor or other health worker that you have high blood pressure or hypertension?’(14), those that responded ‘Yes’ were defined as ‘Aware’ of their hypertension and the ‘No’ group were defined as ‘Unaware’ if they had elevated blood pressure. The Uncontrolled category included hypertensive individuals who were either ‘Unaware’ or those who were ‘Aware’ but not controlled (i.e., had elevated blood pressure at the time of survey). The ‘Controlled’ category was defined as individuals who were ‘Aware’ but did not have elevated blood pressure at the time of survey.’

In our results we have focused the text to the results of spousal concordance for hypertension control, to avoid confusion with hypertension awareness. The low rates of hypertension awareness in Namibia are now raised in the discussion section, with reference to the prevalence of hypertension awareness reported in the DHS final report (page 23). 

‘Yuyun et al. reviewed articles covering the prevalence of cardiovascular diseases (CVD) in SSA from January 1990 to March 2019 and reported that over 60% of hypertensive adults (>18 years old) were unaware of their condition. (31) The low rates of CVD awareness in SSA were attributed to insufficient health care infrastructure and lack of resource allocation towards NCDs. Low rates of awareness are mirrored in the Namibia DHS final report with 49% of hypertensive females and 61% of hypertensive males being unaware that they had elevated blood pressure.(14)’

• Even though secondary data was used, the author has to explain the following points in method part deeply: The study setting? The setting characteristics?, which population data set was used in this analysis?, how many of them fulfilled the inclusion criteria, how many of them were removed/dropped? (492 couples for HTN prevalence Vs 121 for HTN control), what study design was applied ?, how the study subjects were recruited? (all stages of sampling need to be explained in detail), what type of weighting was applied? And why?, how the data quality was assured in NDHS?, how missed variable were managed?, why a Multilevel LR model was applied?, how much was the cluster correlation level (within-cluster correlation)?, how you measured?, at what level of intra-cluster correlation the multilevel analysis is recommended?, what individual and community level factors 

were considered, how you assessed your model fitness? (The steps of model building have to be explained clearly). Generally, the method section is poor in mentioning above issues. Hence, the authors must incorporate these points seriously.

Authors’ response: Thank you for your feedback on the methods section, we have now expanded on the above comments within our methods section, under the subheadings ‘Study Setting, Sample Design and Weight, Questionnaires, Sample Selection, Outcome variables, Confounding (Hypertension Risk Factor) Variables and Data Analysis’, pages 5-7.

• Discussion need to be sequenced based on the order of objectives of this study.

Authors’ response: Thank you for raising this, we have now included to subheadings within our discussion to highlight the research question we are discussing, and these are in sequential order. 

• The discussion is shallow especially for the factors obtained for the HTN. The possible explanations for each variable were not discussed deeply. So, re-write it. 

Authors’ response: Thank you for this comment, we agree that a deeper discussion of possible explanations for each variable was required. We have added discussion of the significant factors: age, BMI, diabetes and residence with reference to past hypertension literature, page 17. Due to space constraints, we have emphasised how our findings are consistent with direction for known risk factors in other studies, page 21/22. Additional text now the discussion… ‘Our findings for significant hypertension risk factors among these Namibian couples are generally consistent with previous literature from other parts of the world.(15, 16, 18, 19) Older age is a widely recognised risk factor for hypertension, this relationship is largely associated with structural changes within arteries as well as calcification over time.(16, 18) A 2007 systematic review of 25 studies across 10 SSA countries reported that urban residence and older age are the most significant determinants of higher hypertension prevalence.(29)

In addition to being an independent risk factor for NCDs, high BMI (≥30 kg/m2) has repeatedly been associated with increased odds of hypertension.(15, 19, 27) Obesity has also been shown to be a risk factor with high spousal concordance. (30) Individuals living with both diabetes and hypertension is another common pattern of comorbidity.(21) Diabetes is, therefore, a significant predictor of hypertension in many studies, including the 2013 DHS in which females with diabetes were more than twice as likely to be hypertensive (OR 2.23 CI 1.40-3.40) than females without diabetes.(15, 22) The shared disease mechanisms and primary risk factors, such as obesity, mean that both diabetes and hypertension can be viewed to have a causal relationship with the other.(21)

Smoking was not a significant risk factor in our study and whilst hypertension and smoking status are risk factors for cardiovascular disease, the influence of smoking on hypertension status is unclear.(23) In contrast to findings, a prospective cohort study of 28,236 American women, found that the risk of hypertension increases in women who smoke more than 15 cigarettes a day (aOR 1.11 (CI 1.03–1.21)) compared to those who have never smoked.(24)’ 

• In discussion, you only tried to discuss those variables for females HTN. What about those factors for males’ HTN?

Authors’ response: We have now also added a sentence on the factors that remained significant for male hypertension, paragraph 2 of page 21. ‘Our final model for male hypertension found individual age and individual BMI were significantly associated with increased odds of hypertension but smoking status was not.’

• In conclusion section, the authors have to make sure that all mentioned recommendations considered all identified factors.

Authors’ response: Thank you for this comment, we have now included how current Namibian policy addresses identified factors within the conclusion. Page 27- ‘Current Namibian policy has listed actions to reduce four behavioural risk factors for NCDs.(33) Despite no reference to hypertension specific interventions, the actions listed within the plan address significant hypertension risk factors seen in this study such as diabetes, high BMI and lack of education.(33)’ We have also expanded on the current Namibian NCD policy within our discussion, on page 18/19 – ‘In terms of the prevention and control of hypertension in Namibia, there is minimal reference to hypertension screening and management within the Ministry of Health’s (MoH) NCD plan for 2017/18 – 2021/22.(33) Four behavioural risk factors for NCDs were recognised within the plan: ‘use of tobacco products, harmful use of alcohol, physical inactivity and unhealthy diets’.(33) Nine targets were set including ‘Halt the rise in obesity and Diabetes Mellitus by 2022’ and ‘A 15% relative reduction in prevalence of raised blood pressure and/or contain the prevalence of raised blood pressure by 2022; and a 25% relative reduction by 2025’.’ and ‘The Namibian National Health Policy Framework recognises the rising levels of NCDs and lists action points which include surveillance of NCD risk factors, institutionalization of NCD screening and ‘strengthening health promotion through behavioural change communication, including community dialogue’. (34)’

• Did the researcher answer the research question? Which was the couple’s concordance in HTN prevalence and HTN control? I do not think so. How can we measure either the concordance exists or not? For me the researcher simply assessed the prevalence of HTN and associated factors for both sexes (males and females), and even though I do not agreed with the measurement of the variable HTN control status, the control status and factors were also assessed. I need clear explanation on these issue, and the author has to make the findings of this study more easily readable for the readers of this document.

Authors’ response: Given all of the above changes we hope that this reviewer has clearer that we have answered the research questions posed in this paper.

• The topics for tables and figures have to be made self-explanatory and the fig. topic needs to be put at the appropriate place, preferably at the bottom of figures.

Authors’ response: Table titles have been edited to be more self-explanatory and Figure 1 has been moved to follow the paragraph where it is first referenced.

• For web page references, the URL, Access date and citation dates have to be incorporated.

Authors’ response: - NLM reference format has been used and web page references have been edited accordingly.

• English editing is also highly needed.

Authors’ response: We have revised text in places for clarity and corrected typos. 

Thank you again for all your suggestions, we believe the changes have strengthened our paper.

Yours sincerely,

Alice Weare

---

## [Decision Letter · Decision Letter 1]

6 Jun 2023

PONE-D-22-17747R1The prevalence of hypertension and hypertension control among married Namibian couples.PLOS ONE

Dear Dr. Weare,

Thank you for submitting your manuscript to PLOS ONE. After careful consideration, we feel that it has merit but does not fully meet PLOS ONE’s publication criteria as it currently stands. Therefore, we invite you to submit a revised version of the manuscript that addresses the points raised during the review process.

We look forward to receiving your revised manuscript.

Kind regards,

Melkamu Merid Mengesha, MPH

Academic Editor

PLOS ONE

Journal Requirements:

Additional Editor Comments:

It is of great pleasure that the authors have thoroughly addressed reviewer's comments in their revised submission.

Just a minor comment:

The authors should identify specific confounding variables than putting an equivalence between hypertension risk factors versus confounding factors.

Also add who collected the data and efforts taken to maintain data quality.

Does the definition for obesity line 145 inclusive of BMI=30?

In figure 1, exclusion of 605 couples is from 1249 couples not from the 644 couples. as it currently stands, the exclusion seems from the 644 couples, and this should get corrected.

Reviewers' comments:

Reviewer's Responses to Questions

**Comments to the Author**

1. If the authors have adequately addressed your comments raised in a previous round of review and you feel that this manuscript is now acceptable for publication, you may indicate that here to bypass the “Comments to the Author” section, enter your conflict of interest statement in the “Confidential to Editor” section, and submit your "Accept" recommendation.

Reviewer #1: All comments have been addressed

2. Is the manuscript technically sound, and do the data support the conclusions?

Reviewer #1: Yes

3. Has the statistical analysis been performed appropriately and rigorously? 

Reviewer #1: Yes

4. Have the authors made all data underlying the findings in their manuscript fully available?

Reviewer #1: Yes

5. Is the manuscript presented in an intelligible fashion and written in standard English?

Reviewer #1: Yes

6. Review Comments to the Author

Reviewer #1: The authors have addressed all my comments and significantly improved the manuscript. No further comments.

7. PLOS authors have the option to publish the peer review history of their article (what does this mean?). If published, this will include your full peer review and any attached files.

Reviewer #1: No

While revising your submission, please upload your figure files to the Preflight Analysis and Conversion Engine (PACE) digital diagnostic tool, https://pacev2.apexcovantage.com/. PACE helps ensure that figures meet PLOS requirements. To use PACE, you must first register as a user. Registration is free. Then, login and navigate to the UPLOAD tab, where you will find detailed instructions on how to use the tool. If you encounter any issues or have any questions when using PACE, please email PLOS at figures@plos.org. Please note that Supporting Information files do not need this step.<quillbot-extension-portal></quillbot-extension-portal>

---

## [Author Response · Author response to Decision Letter 1]

20 Jul 2023

Dear PLOS ONE editor and reviewers,

Thank you for taking the time to read and review our revised submission ‘The Prevalence of Hypertension and Hypertension Control Among Married Namibian Couples’. We are pleased that the reviewers felt all their comments have been addressed. We have addressed each of the editor comments and a point-by-point response in italics is included below.

Additional Editor Comments:

It is of great pleasure that the authors have thoroughly addressed reviewer's comments in their revised submission.

Just a minor comment:

1. The authors should identify specific confounding variables than putting an equivalence between hypertension risk factors versus confounding factors.

Authors’ response: We kept all significant factors (age, obesity, education, diabetes status, current smoking status, household wealth and residence) in the models as an indicator that they are hypertension risk factors. We did not formally assess whether they were confounders (changing the association of interest), therefore we have reviewed our references to potential confounders and amended the text for clarity. 

2. Also add who collected the data and efforts taken to maintain data quality.

Authors’ response: As this was secondary analysis of the Namibian DHS we have included additional detail regarding data collection and quality assurance from the DHS survey final report, line 73-75.

3. Does the definition for obesity line 145 inclusive of BMI=30?

Authors’ response: The definition of obesity has been corrected to show inclusive of BMI=30, i.e. BMI≥30 (line 147).

4. In figure 1, exclusion of 605 couples is from 1249 couples not from the 644 couples. as it currently stands, the exclusion seems from the 644 couples, and this should get corrected.

Authors’ response: The format of figure one has been corrected to clearly show the point at which exclusions took place. 

Thank you again for all your suggestions, we believe the changes have strengthened our paper.

Yours sincerely,

Alice Weare

---

## [Editor Report · Decision Letter 2]

27 Jul 2023

The prevalence of hypertension and hypertension control among married Namibian couples.

PONE-D-22-17747R2

Dear Dr. Weare,

We’re pleased to inform you that your manuscript has been judged scientifically suitable for publication and will be formally accepted for publication once it meets all outstanding technical requirements.

Within one week, you’ll receive an e-mail detailing the required amendments. When these have been addressed, you’ll receive a formal acceptance letter, and your manuscript will be scheduled for publication.

An invoice for payment will follow shortly after the formal acceptance. To ensure an efficient process, please log into Editorial Manager at http://www.editorialmanager.com/pone/, click the 'Update My Information' link at the top of the page, and double check that your user information is up to date. If you have any billing related questions, please contact our Author Billing department directly at authorbilling@plos.org.

Kind regards,

Melkamu Merid Mengesha, MPH

Academic Editor

PLOS ONE

Additional Editor Comments (optional):

Reviewers' comments:

<quillbot-extension-portal></quillbot-extension-portal>

---

## [Editor Report · Acceptance letter]

31 Jul 2023

PONE-D-22-17747R2 

The prevalence of hypertension and hypertension control among married Namibian couples. 

Dear Dr. Weare:

I'm pleased to inform you that your manuscript has been deemed suitable for publication in PLOS ONE. Congratulations! Your manuscript is now with our production department. 

Kind regards, 

on behalf of

Mr. Melkamu Merid Mengesha 

Academic Editor

PLOS ONE